# Drug Design Targeting the Muscarinic Receptors and the Implications in Central Nervous System Disorders

**DOI:** 10.3390/biomedicines10020398

**Published:** 2022-02-07

**Authors:** Chad R. Johnson, Brian D. Kangas, Emily M. Jutkiewicz, Jack Bergman, Andrew Coop

**Affiliations:** 1Department of Pharmaceutical Sciences, University of Maryland School of Pharmacy, Pharmacy Hall North Room 623, 20 N. Pine St., Baltimore, MD 21202, USA; acoop@rx.umaryland.edu; 2Behavioral Biology Program, McLean Hospital, Harvard Medical School, 115 Mill St., Belmont, MA 02478, USA; bkangas@mclean.harvard.edu (B.D.K.); jack_bergman@hms.harvard.edu (J.B.); 3Department of Pharmacology, University of Michigan, 1150 W. Medical Center Dr., Ann Arbor, MI 48109, USA; ejutkiew@umich.edu

**Keywords:** Alzheimer’s disease, schizophrenia, depression, major depressive disorder, drug design, muscarinic receptors, muscarinic agonist, muscarinic antagonist, positive allosteric modulator

## Abstract

There is substantial evidence that cholinergic system function impairment plays a significant role in many central nervous system (CNS) disorders. During the past three decades, muscarinic receptors (mAChRs) have been implicated in various pathologies and have been prominent targets of drug-design efforts. However, due to the high sequence homology of the orthosteric binding site, many drug candidates resulted in limited clinical success. Although several advances in treating peripheral pathologies have been achieved, targeting CNS pathologies remains challenging for researchers. Nevertheless, significant progress has been made in recent years to develop functionally selective orthosteric and allosteric ligands targeting the mAChRs with limited side effect profiles. This review highlights past efforts and focuses on recent advances in drug design targeting these receptors for Alzheimer’s disease (AD), schizophrenia (SZ), and depression.

## 1. Introduction

With nearly 800 human genes coding for them, G protein-coupled receptors (GPCRs) are a large family of plasma membrane proteins that are highly conserved throughout evolution [1]. The successful development of multicellular organisms was, in part, dependent on the evolution of these proteins, which serve to translate extracellular signals to intracellular functions, thereby facilitating communications between cells and their environment [2]. GPCRs are widely present in most life forms, including bacteria, fungi, and higher animals, and are involved in nearly all aspects of animal physiology [3]. They facilitate several cell signaling transduction cascades that regulate various critical cellular processes and have been implicated in a wide range of biological functions, including behavior, cognition, immune response, mood, olfaction, blood pressure regulation, and taste, among others [4,5].

The cell membrane location and diversity of tissue expression of GPCRs make them ideal targets for drug discovery. Out of the 219 new molecular entities (NMEs) approved by the US Food and Drug Administration (FDA) from 2005–2014, 54 (~25%) targeted GPCRs [6]. Approximately 80 known GPCRs are currently targeted by approved therapies (~10% of the known coded receptors), leaving considerable room to discover novel compounds. However, this task has been complicated by the (1) commonality of ligands targeting subfamilies of GPCRs; (2) the high homology among ligand binding sites; and (3) difficulty obtaining 3-dimensional structures of the receptors in both their active/inactive states. 

The muscarinic acetylcholine receptors (mAChRs) are a family of five closely related class A GPCRs (M_1_–M_5_) encoded by the *CHRM1-CHRM5* genes, which have been highly conserved throughout evolution [7]. In addition to the key role of mAChRs in the autonomic nervous system, they further regulate several essential central nervous system functions, including those involved in cognition as well as motor and sensory function. In turn, disturbances in mAChR function have been implicated in Alzheimer’s disease (AD), Parkinson’s disease, depression, and schizophrenia (SZ).

The first mAChR structure (M_2_) was promulgated by Haga et al. [8] in 2012, serving as a significant breakthrough in structure determination for the mAChRs. The receptor was crystallized in its inactive conformation bound to the inverse agonist 3-Quinuclidinyl benzilate (QNB), providing a snapshot of the orthosteric binding pocket and identifying potential allosteric sites in the extracellular loops (ECLs, near the aromatic cap). The M_3_ receptor structure bound to tiotropium (Spiriva), a treatment for asthma/COPD, was similar to M_2_, including its intracellular and extracellular loops [9]. In 2016, crystallization of the M_1, 4_ receptors revealed only slight differences in the orthosteric sites but confirmed larger differences in the allosteric binding sites on the ECLs, potentially offering an opportunity to achieve mAChR subtype selectivity [10]. Along these lines, mAChR (*CHRM1-CHRM5*) knockout (KO) mice [7] have provided valuable insights into the physiological and pathophysiological roles of the individual subtypes (reviewed in Wess et al. [11]).

Thus, the cholinergic system offers exciting opportunities for researchers in academia and industry alike, and in particular, the cholinergic system continues to provide fertile ground for drug design and discovery. The present review adopts a chronological approach to highlight the historical developments (e.g., the cholinergic hypothesis and Merck’s early efforts designing mAChR agonists) and shifts to more recent drug development efforts and design (e.g., allosteric modulators and select orthosteric compounds), specifically for AD, SZ, and major depressive disorder (MDD). In addition, although not exhaustive, some of the most promising compounds and chemical scaffolds will be discussed to illustrate their potential value for treating these pathologies. 

## 2. Drug Design Targeting the mAChRs: Alzheimer’s Disease

### 2.1. Orthosteric Agonists

Active research efforts in the 1960s–1980s helped to develop an understanding of neurotransmitters’ physiological roles, which ultimately led to the consideration that the altered function of neurotransmitter pathways was associated with several CNS disorders [12]. This was validated via a tissue analysis of samples from the brains of postmortem AD patients, which revealed a cholinergic projection deficit from the basal forebrain neuronal population (nucleus basalis magnocellularis of Meynert) to the cortex and hippocampus. Furthermore, the activity of choline acetyltransferase (ChAT), the enzyme primarily responsible for the synthesis of acetylcholine and a marker of cholinergic neurons/synapses, was found to be significantly decreased in the cortex/hippocampus of AD patients [13,14]. Finally, depolarization-induced acetylcholine release and choline uptake in nerve terminals were reduced in these identical tissue specimens [15,16]. Based on these observations, the cholinergic model of impairment was first presented in a review by Bartus et al. [17], which concluded that the critical event in the pathology of AD was the degeneration of the cholinergic connection from the nucleus of Meynert to the cortex and hippocampus [18].

AD is the most common neurodegenerative disorder that affects the elderly, resulting in memory loss and severe cognitive dysfunction [19]. The disease itself is complex but can be generally characterized by the following two main events in the brain: (1) aggregates of amyloid plaques (primarily composed of amyloid-β peptide [Aβ]), and (2) neurofibrillary tangles formed via hyper-phosphorylated tau proteins [20]. Aβ is generated by proteolytic cleavage of amyloid precursor protein (APP), which can be processed in two ways. A majority of APP undergoes cleavage by consecutive α (or γ)-sectretases yielding soluble, non-pathogenic APPα. Conversely, APP can be sequentially cleaved by β (and γ)-secretases to yield soluble APPβ and Aβ (neurotoxic) [21]. Aβ oligomer formation appears to affect hippocampal synaptic transmission, which is thought to be related to changes in synaptophysin expression and severely diminished ChaT transcription, leading to decreased ChAt activity and the steady progression of dementia [22].

In early efforts to improve cholinergic transmission and enhance cognitive function of AD patients, acetylcholinesterase (AChE) inhibitors (e.g., tacrine, physostigmine, donepezil, galantamine, rivastigmine) and directly acting agonists at postsynaptic muscarinic receptors in the cerebral cortex (e.g., arecoline, RS-86, and pilocarpine) (Figure 1) were utilized as a first-line treatment [23]. Of the aforementioned classes of compounds, the AChE inhibitors showed the greatest success in early clinical trials due, in part, to the side effects associated with non-specific mAChR stimulation via direct-acting agonists. While AChE inhibitors are still used today for symptom treatment, their efficacy is modest at best [24] and, thus, the cognitive symptoms of AD (e.g., memory loss, confusion, problems thinking/reasoning, etc.) are poorly managed. Additionally, cardiovascular and gastrointestinal side effects are often observed with these treatments (thought to be mediated by peripherally located ACh receptors), and beneficial effects tend to dissipate with tolerance [25,26]. 

Merck’s early efforts to improve the clinical profile of arecoline, a weak partial agonist in the cortex compared to ACh, involved replacing the ester group (pink, Figure 1) with the metabolically stable 3-methyl-1,2,4-oxadiazole (red/green, Figure 1) bioisostere. The arecoline-based oxadiazoles (1, 2) demonstrated greater potency than arecoline itself but offered only minimal efficacy improvements, leading Merck to develop the quinuclidine (1-azabicyclo[2.2.2]octane) and azanorbornane (1-azabicyclo[2.2.1]heptane)-methyl and amino oxadiazoles (3, 4, 5a/b, 6a/b) [27]. Both showed marked improvements in potency and efficacy over the arecoline series. The 3-exo configuration of the azanorbornanes displayed superior potency and efficacy among the series, attributable to a lower steric demand on the binding surface within the active site. The amino oxadiazole (red/blue, Figure 1) proved to have the highest potency and predicted efficacy, presumably due to the oxadiazole ring’s increased hydrogen bonding capabilities. These early efforts led to the creation of a library of mAChR agonists over the next few decades that permitted elucidation of structure-activity relationships (SAR) among orthosteric ligands. 

It was demonstrated that increasing the conformational flexibility and bulk of the azacyclic ring reduced affinity compared to the more rigid arecoline, azanorbornane, and quinuclidine cores. Additionally, increasing the pKa of the hindered amine (i.e., enhancing its ability to become protonated once in the CNS, and hence mimic the quaternary nitrogen of ACh) enhanced the affinity for the mAChRs [27]. Therefore, Merck chemists and others shifted their focus to arecoline-, quinuclidine-, and azanorbornane-core structures in all subsequent studies. While successful at developing some of the most potent non-quaternary agonists to date, the novel compounds provided little therapeutic benefit, due to dose-limiting cholinergic side effects that likely reflected their lack of subtype selectivity. 

Neurochemical examination of brain material from AD patients showed a loss of the presynaptic marker enzyme (e.g., loss of synaptic terminals), choline acetyltransferase, and M_2_- but not the postsynaptic M_1_-mAChRs in various brain regions [28,29,30]. Hence, drug development efforts shifted in the early 1990s to the generation of functionally selective M_1_ agonists. 

As is known, M_1_ is the most abundant of the five mAChRs in brain regions (particularly in the prefrontal cortex and hippocampus) involved in cognitive processes [31,32]. Pharmacological blockade [33,34] or genetic deletion [35] of M_1_ produces significant learning and memory disturbances, including deficits in social interaction, social discrimination, and working memory (i.e., radial arm maze test). As several groups have implicated the role of central mAChRs in cognitive processes [36,37,38] as well as in APP processing [39,40], it was reasonable to ask whether selective activation of M_1_ could enhance cognition and reverse learning and memory disturbances. Sauerberg et al. [41], and Ward et al. [42] were among the first to design M_1_ selective compounds of the 3-(1,2,5-thiadiazolyl)-1,2,5,6-tetrahydro-1-methylpyridine and 3-pyrazinyl-1,2,5,6-tetrahydro-1-methylpyridine type. The hexyloxythiadiazole derivative, known as Xanomeline (thiadiazole moiety in blue, Figure 2), showed early promise as a potential treatment for AD. Binding studies showed that it was a subtype-selective M_1_ receptor agonist; however, other in vitro/in vivo functional studies suggest that the compound is better classified as a (slightly) subtype-selective M_1_/M_4_ agonist. 

Xanomeline was found to improve psychosis and behavioral disturbances in those suffering from AD [43] and also had positive effects in schizophrenic patients [44]. Unfortunately, the drug’s clinical use was limited by its side effect profile that included salivation, sweating, and gastrointestinal distress, all of which are likely attributed to the nonselective stimulation of other mAChRs (primarily M_3_ receptors) [45,46]. Although GI side effects were the culprit of the high drop-out rate within clinical trials [47,48], the compound is now being co-administered with trospium chloride (a generic drug for overactive bladder) to counteract peripheral autonomic effects that caused earlier trials to cease. KarXT (Xanomeline tartrate and trospium chloride) is now in Phase 2 trials (NCT02831231) to treat the cognitive symptoms of SZ, psychosis, and agitation related to dementia, including AD, and as a non-opioid based therapy for forms of post-operative inflammatory and neuropathic pain (https://karunatx.com/programs/; Accessed 29 November 2021). 

Efforts to develop M_1_-selective ligands and lower efficacy partial agonists to prevent cholinergic side effects continued for several years but with limited success [49]. Tetrazole (red, Figure 2)/1,2,3-triazole scaffolds [50], oxime ether functionality (orange, Figure 2) [51,52], 4-thiazolidinone (gold, Figure 2), and ether linkages directly to functionalized pyrazine rings (green, Figure 2) were designed in an attempt to replace the 1,2,4-oxadiazole ring; ultimately leading to the development of the first generation of compounds (Tazomeline [53], Alvameline [54], Cevimeline [55], Milameline [56], NGX-267 [57], Talsaclidine [58], Sabcomeline [59], and WAY-132983 [60], Figure 2). All failed at some stage in clinical development.

Despite recent efforts shifting to the design of highly selective allosteric modulators, Heptares Therapeutics (GPCR focused biotech, a subsidiary of Sosei) designed two highly selective M_1_ orthosteric agonists in phase 1 clinical development. Their proprietary approach incorporates the construction of stabilized mAChRs in their agonist conformations, followed by introducing mutations to allow for an improved stability of the mAChR in various conditions. These stabilized proteins permit crystallization with a wide selection of in-house and competitor ligands. The crystal structures generated in combination with site-directed mutagenesis allowed for the development of ligand binding models to the M_1_ mAChR, to which virtual screening could be applied to identify potential leads (StaR^®^ technology) [61]. In 2016, Allergan acquired rights to develop further the muscarinic drug pipeline with the novel piperidinyl-azepine HTL-9936 (pink, Figure 1) and HTL-18318 (structure not published) as the lead candidates. 

Additionally, HTL-9936 is a moderately efficacious partial agonist (45–75% efficacy compared to ACh, EC_50_~100 nM) at the M_1_ mAChR, displaying a modest selectivity (~7-fold) over M_2_ and two-fold selectivity for M_1_ over M_4_. Moreover, the molecule did not display functional agonist activity for human M_2_ and M_3_ mAChRs. Consistent with its in vitro profile, this compound was able to reverse a scopolamine-induced deficit in a passive avoidance task in rats and showed efficacy in rodent novel recognition and aged Beagle dog object retrieval tasks [62]. Heptares announced positive phase 1b results (no adverse events or GI distress) in early 2016 (NCT02291783); however, more recent data has not yet been published. As noted by Felder [63], brain penetration was predicted to be low compared to plasma, and indeed, as indicated by a Kp_u,u_~0.05 (generated by Lilly), brain exposures are anticipated to be approximately 20-fold lower than plasma values. In general, Kp_u,u_ values of ~0.3 or greater are considered to have sufficient access to the CNS, with values greater than 1 representing compounds that freely cross the BBB (for review see Kulkarni et al. [64]. Kp_u,u_ represents the free fraction of the drug in the brain divided by the free fraction of drug in the plasma. The above data were calculated 30 min after oral dosing at 100 mg/kg using measured rat plasma and rat brain fraction unbound. Despite the low Kp_u,u_ value, it is likely sufficient for efficacy due to the high receptor reserve (i.e., activation of a fraction of receptors is sufficient for a full cellular response—see Schrage et al. [65], Rajagopal et al. [66], and Buchwald [67] for review) within the areas of the brain related to cognition. 

Little has been published on HTL-18318, which is also derived from Sosei Heptares’ proprietary StaR technology (detailed above) and has a structure-based drug design. HTL-18318 is a potent M_1_ agonist with negligible activity at M_2_/M_3_ (M_4_/M_5_ not specified) that has undergone a phase I clinical trial for AD in the United States (NCT03456349) and a phase II trial in patients with Lewy body dementia in Japan (JapicCTI-183989). The compound has been investigated in over 300 human subjects (in both the US and Europe), including healthy volunteers and those with mild to moderate AD, with no serious adverse effects up to 28 days post-treatment. In a press release by Sosei Heptares in 2018, their upcoming Phase 2 clinical trial (JapicCTI-183989) had been voluntarily halted due to a single animal toxicology study (non-human primates) investigating dosing levels of this compound over nine months. In their statement, some animals developed tumors, albeit at doses higher than those expected to be used in humans, thereby causing further development (including a planned Phase 2 study for AD) to be suspended [68].

While there was no publication by Sosei Heptares detailing the results of the above-mentioned dosing study, the clinical evaluation of HTL-18318 continued in Europe. In one study, a randomized, double-blind, placebo-controlled study in 40 healthy young adults and 57 healthy elderly subjects showed that the compound [1–35 mg, administered PO] was absorbed rapidly (peak concentration after 1–2 h) and had a half-life of ~12–16 h. More importantly, while approximately 30% of the plasma unbound concentration entered the cerebral spinal fluid, single doses were associated with only mild dose-related adverse effects (hyperhidrosis, slight increases in blood pressure), yet no statistically significant effects on cognitive function [69]. In a follow-up study, treatment with HTL18318 over ten days displayed significant improvements in tests of short-term working memory (n-back test) and learning (Milner maze) (NTR5781) [70]. 

While ligands such as HTL18318 may generate some optimism, drug design targeting the orthosteric site for AD treatment continues to be challenging for academics and industry alike, which can be attributed to the highly homologous binding pocket of the mAChRs. Additionally, as ACh is a permanent cation, it presents a unique problem for designing (and modeling) an orthosteric agonist that effectively penetrates the CNS, while possessing a sufficiently basic moiety to allow for protonation, hence mimicking the cationic nature of ACh. Nevertheless, advances in VS (Virtual Screening)/high throughput screening (HTS) and crystallization techniques have made it possible to target the orthosteric site with some degree of functional selectivity. The essential feature appears to be the delicate balance between brain penetration and partial-agonist-like efficacy profile to minimize peripheral AEs and M_1_ mediated adverse effects. While not yet applied in clinical trials, the most recent advancement has come from Eli Lilly in late 2018 in the form of SPP1 (Figure 1), a selective M_1_ partial agonist based on a spiro-piperidine core (similar to AZD-6088), which showed >100-fold functional selectivity versus M_2_/M_3_ (a clear improvement over HTL-9936) [71,72]. Significantly, these successful efforts demonstrate that high sequence homology need not preclude the development of orthosteric compounds and indicate that the field remains open for continued exploration.

### 2.2. Allosteric Modulators 

Since the discovery of gallamine as an allosteric modulator by Clark and Mitchelson [69], considerable effort has been directed to understanding its mechanism. HTS capabilities, high-resolution crystal structures (with and without allosteric modulators bound), and powerful computational simulation capabilities have paved the way for identifying several diverse chemical scaffolds that display high functional selectivity. M_1_ continues to be the target of choice for AD due to its alleged vital role in cholinergic deficit, cognitive dysfunction, and tau/Aβ pathologies [39,73,74]. 

The difficulty of obtaining a high degree of functional selectivity targeting the orthosteric site has led researchers to focus on allosteric ligands, which target less conserved receptor regions (e.g., ECLs). Mechanistically this may reflect several advantageous considerations. For example, modulators can be identified with higher affinity for one subtype over others because of lower sequence conservation in allosteric sites. Additionally, subtype selectivity can also arise from cooperativity or, in the case of ‘absolute subtype selectivity’, both cooperativity and affinity [75]. Furthermore, an allosteric effect only arises when the endogenous ligand (ACh) is present and, therefore, better mimics the physiological regulation of receptor activity (e.g., enhances cholinergic tone) [8]. Finally, allosteric actions are saturable (exhibit a “ceiling” effect) [76], which allows for a wider safety margin (e.g., less adverse side-effect profile). 

The first and most often studied allosteric modulators of the mAChRs were the neuromuscular blocking agents (gallamine/alcuronium) and a series of alkyl-bis-quaternary ammonium compounds (Obidoxime/C7/3-phth/W84, pink, Figure 3) [77,78]. Alcuronium was found to be the first allosteric enhancer of the binding of an orthosteric mAChR ligand [79,80], and was shown to bind at the same site as gallamine and the alkyl-bis-quaternary ammonium compounds [81]. A second allosteric site was identified shortly thereafter through the observation that derivatives of staurosporine (a nonselective kinase inhibitor) exhibited positive, negative, or neutral cooperativity but did not bind to the already known allosteric site (e.g., that of gallamine and brucine, Figure 3) [82].

Furthermore, McN-A-343 [83,84,85], AC-42 (Acadia Pharmaceuticals, Inc., butyl piperidine core, blue, Figure 3) [84], AC-260584 [86,87,88,89], 77-LH-28-1 (GSK, Figure 3) [90], and TBPB (bis-piperidinyl core, green, Figure 3) [40,91], were among the first generation of allosteric modulators (for review see Conn et al. [92] and Heinrich et al. [60]). However, they suffered drawbacks from lack of selectivity and efficacy in vivo, off-target effects, intrinsic activity, and unsuitable PK properties, which prevented their further development for clinical use. 

Merck’s first prototypical M_1_ PAM, benzyl quinolone carboxylic acid (BQCA, and derivatives, Figure 4), was a highly selective potentiator M_1_ receptor while demonstrating no action at any other subtype (up to 100 µM). It reduced scopolamine-induced memory deficits in contextual fear conditioning (CFC) assays, restored discrimination reversal learning in a transgenic mouse model of AD, regulated amyloidogenic APP processing in vitro, and reversed amphetamine-induced hyperlocomotion, which is an in vivo assay of antipsychotic efficacy) [92,93]. While the compound’s low efficacy precluded its ascension to clinical trials, the scaffold was optimized for safety and bioavailability, which led to more successful derivatives [94,95,96,97,98,99].

Additionally, MK-7622 (Merck) represented a first-in-class allosteric modulator optimized from the BQCA core scaffold. While BQCA (and subsequently 1-((4-cyano-4-(pyridine-2-yl)piperidin-1-yl)methyl-4-oxo-4H-quinolizine-3-carboxylic acid, PQCA) was useful in probing the pharmacological effects of allosteric modulation, low CNS penetration, and rapid efflux out of the CNS via the phospho-glycoprotein (P-gp) transporter resulted in poor efficacy. Replacing the carboxylic acid group of BQCA/PQCA through introducing a pyrimidinone ring (blue, Figure 4) appeared to offer a balance between low P-gp efflux and good oral bioavailability, and clinical efficacy. P-gp is likely the most well-known of the ABC (ATP-binding cassette) superfamily that acts as an efflux pump with broad substrate specificity—albeit it has been shown to preferentially extrude large, hydrophobic, positively charged molecules [100,101,102]. It is expressed in a variety of tissues (e.g., GI tract, kidney, liver, endothelium, etc.), including a continuous single layer of brain capillary endothelial cells with tight junctions [103]. There is a clear relationship between P-gp efflux and brain exposure, therefore, a P-gp screening assay is imperative for CNS drug discovery programs. The brain-to-plasma concentration ratio (Kp_,brain_) and the unbound brain-to-plasma concentration ratio (Kp_u,u_, mentioned above) are most commonly used as in vivo PK parameters to assess brain penetration of a compound. While both in vitro and in vivo assays are used in drug discovery, there is often a lack of correlation between the two, likely due to differential expression of P-gp in different cell lines. Therefore, it is preferable to calculate these PK parameters using in vivo experiments in rodents—despite the inherent limitation that differences in P-gp expression or in transport potentials of substrates across species likely exist [104]

Furthermore, MK-7622 was shown to reverse scopolamine’s effects at 3 mg/kg in a mouse contextual fear conditioning assay [100]. Merck began a Phase 2 trial in 2013 to evaluate the compound’s tolerability and efficacy as a symptomatic adjunct to donepezil therapy. The trial was expected to enroll 830 patients but was terminated in 2016 after enrolling only 240 participants. A futility analysis indicated that 12 weeks of treatment with MK-7622, given in combination with AChE inhibitors, did not improve cognition in patients with mild/moderate AD (NCT01852110) [105]. These disappointing results were suggested to have reflected the compound’s ability to activate M_1_ mAChRs in the absence of ACh (intrinsic activity), thereby disrupting prefrontal cortex (PFC) function [106,107].

Davoren et al. [32] (Pfizer) disclosed a series of PQCA derivatives based on the truncation of the quinolizinone core (green, Figure 4) and installation of the 5-methyl group on the pyridine ring to retain the optimal geometry of the benzyl group (V-shaped, or “bent,” conformation) and intramolecular hydrogen bond. After optimization, they arrived at PF-06767832, which displayed ~180-fold selectivity for M_1_ over M_2–5_ with no observed off-target effects and high cooperativity with ACh (αβ value = 126, intracellular Ca^2+^ assays). In vivo, this molecule reversed amphetamine-stimulated locomotor activity and ameliorated scopolamine-induced deficits in spatial learning and memory in the Morris water maze [32]. At low concentrations, it acted as an M_1_ PAM (EC_50_ = 60 nM) but, at higher concentrations, it displayed intrinsic agonist activity, which led to adverse cholinergic AEs that, depending on dosage and regimen, progressed from GI disturbance to generalized convulsions [32,106,107]. These negative results led to the development of PF-068274430, a covalently constrained (γ-lactam, purple, Figure 4) derivative of PF-06767832, that was reported to have a lower intrinsic agonist activity and cooperativity score (αβ = 70, intracellular Ca^2+^ assay). Nevertheless, this compound also produced AEs at elevated doses, precluding its development in the clinic. 

Other companies, namely Asceneuron (a subsidiary of Merck, WO2014198808), Roche (WO2015028483), and Takeda (WO2016208775, TAK-071, Figure 4), also have projects dedicated to M_1_ PAM development, with TAK-071 being the only compound to have reached clinical trials. Research groups, including those led by Jeffrey Conn and Craig Lindsley at Vanderbilt, were among the first to report the development of highly selective M_1_ PAMs [92], discoveries that led to the development of the Warren Center for Neuroscience Drug development Discovery (WCNDD, wcndd.com; Accessed 01 December 2021) and a vast library of allosteric agonists, M_1_ PAMs, and antagonists (see review by Kuduk and Beshore [108] and, more recently, Wold et al. [109]). 

More recent efforts include novel M_1_ PAMs, VU0453595 (6,7-dihydro-5*H*-pyrrolo[3,4-*b*]pyridine-5-one core, orange, Figure 4) and VU0550164 (isoindolin-1-one core, light green, Figure 4), which were shown to lack constitutive agonist activity in cell lines and maintain activity dependence of M_1_ activation in the PFC. Moreover, unlike MK-7622, VU0453595 did not induce behavioral convulsions, even at doses well above that required to improve cognitive function [110]. These pivotal findings support the conclusion that the in vivo cognition-enhancing efficacy of M_1_ PAMs without intrinsic activity is possible and, additionally, may provide a more favorable side-effect profile. Since these findings, further optimization of the PK/PD properties of the core scaffolds has further improved levels of CNS penetration (K_p_s~0.30–3.1, see Panarese et al. [111]).

Moreover, VU319 (structure not published) is perhaps the most successful novel M_1_ PAM to emerge from Vanderbilt’s efforts. The compound underwent a phase 1 clinical trial in 2017 to establish safety and tolerability and to characterize its PK/PD profile (NCT03220295). Currently, no results have been posted for this trial. However, results were presented at the 2020 Alzheimer’s Association International Conference (AAIC) annual meeting, indicating that VU319 demonstrated a safe profile and, based on cognitive performance and EEG measures of event-related potentials, enhanced cholinergic function [112,113]. The group also reported an absence of documented dose-limiting cholinergic adverse effects. Although a firm evaluation can only be provided once the results are published, these reported findings are consistent with the idea that VU319 displays high functional selectivity for the M_1_ mAChR (and low intrinsic activity). A second Phase 1 trial was initiated in Fall 2019 (NCT04051801) to establish the maximum tolerable dose of VU319 and, presuming its profile remains favorable, it is likely that VU319 soon will be evaluated in a multiple ascending dose study as well as a Phase 2a proof of concept trial in patients with mild cognitive impairment. The latter studies will be especially interesting, as they will speak directly to the clinical value of allosteric modulation of M_1_ receptors. 

In 2015–2016, Anavex Life Sciences, a biopharmaceutical company developing drug candidates for AD and other CNS diseases, unveiled both ANAVEX 2–73 (blarcamesine) [114] and ANAVEX 3–71 (previously known as AF170B) (Figure 4). 2–73 is a novel aminotetrahydrofuran (pink, Figure 4) derivative that is a mixed agonist of the chaperone protein sigma-1 (IC_50_ = 0.86 μM) and the M_1–4_ receptors (IC_50_ = 3.3–5.2 μM) [115]. The compound (0.01–3.0 mg/kg i.p.) was shown to alleviate scopolamine and dizocilpine-induced learning impairments and reverse learning deficits (300 μg/kg) in mice injected with synthetic Aβ (Aβ_25–35_) oligomer (likely via inhibition of Aβ_25–35_-induced oxidative stress in the hippocampus) [114,115,116]. Subsequent studies by Lahmy et al. [115,117] suggested that the compound may block tau hyperphosphorylation and amyloid-β1-42 in an Aβ_25–35_ mouse AD model. On the basis of these findings, a Phase 2b/3 (NCT03790709) clinical trial was undertaken and recently completed the enrollment of patients (*n* = 509) with mild-to-moderate AD to evaluate the effects of ANAVEX 2-73 on cognition and function (ADCS-ADL) after 48 weeks of daily treatment. Although ANAVEX 2-73 was initially designed to treat AD (i.e., cholinergic hypofunction), it is also undergoing a Phase 2 evaluation of its efficacy in treating Parkinson’s disease dementia (i.e., cholinergic hyperfunction) (NCT03774459) and Rett Syndrome (NCT03941444 and NCT04304482, Kaufmann et al. 2019). These latter studies are likely designed to evaluate ANAVEX 2-73′s activity at the sigma-1 receptor. Thus, activation of sigma-1 mechanisms has been shown to result in lower toxic accumulation of misfolded proteins in nerve cells, dysfunction in mitochondria, oxidative stress, and neuroinflammation, all of which are involved in Parkinson’s, AD, and Rett syndrome. Of interest, early results of the PDD trial, presented in 2020, suggested that treatments with 30 or 50 mg daily for 14 weeks resulted in improvements in memory and attention measures compared to placebo. To explore these effects further, the trial currently has an open-label extension [118]. 

With regard to ANAVEX 3-71 (Figure 4), it shares some structural similarities with the early orthosteric agonist NGX267 (Figure 1) but displays a strikingly different pharmacological and binding profile. Similar to 2-73, it is a chiral (*S*-enantiomer, US8673931B2), highly selective mixed sigma-1 (0.25 μM), and M_1_ (41 μM, no agonistic activity at M_2_-M_5_) PAM, with no appreciable off-target activity. Moreover, the compound effectively reversed cognitive deficits induced by trihexyphenidyl (1–30 μg/kg), decreased tau phosphorylation induced by Aβ_25–35_, exerted neuroprotective effects (low μM), and rescued mushroom spines in hippocampal neuron culture from PS1.KI and APP.KI mice [119]. The (S)-enantiomer of this molecule is in Phase 1 trials for safety, tolerability, and PK (NCT04442945). Interestingly, the (R)-enantiomer is inactive up to 10 μg/kg p.o., indicating that the effects of AF710B on cognition are enantioselective. 

Marketed as a highly selective M_1_ PAM with a lower α-value (~199), TAK-071 (Takeda Pharmaceutical Company Ltd, Osaka, Japan), shows some similarities in structure to VU055064 (isoindolin-1-one core, green). The compound was shown to improve scopolamine-induced cognitive deficits in rats (0.3 mg/kg) with a reduced side effect profile (GI distress tolerable up to 10 mg/kg) [120]. Additionally, it induced afterdepolarization in layer V pyramidal neurons and, when combined with AChEIs, it exhibited synergistic effects in reversing scopolamine-induced deficits in the novel object recognition task. The compound entered Phase 1 trials in 2016 to identify a safe and well-tolerated dose (NCT02769065) and shortly thereafter to evaluate its ability to ameliorate cognitive impairments induced by scopolamine (in combination with donepezil) (NCT02918266). Both sets of trials were terminated early on due to indication change, and TAK-071 is currently being evaluated for its effects on falls in patients with Parkinson’s disease (NCT04334317). 

Given the high homology of the orthosteric site, it is not surprising that most recent efforts have shifted to targeting the less conserved allosteric binding regions in the ECLs of the mAChRs. Additionally, PAMs seem to offer an advantage to direct acting agonists by enhancing cholinergic tone via modulating the effects of natural neurotransmitters, and hence better mimics physiological receptor activity, much progress has been made through modeling, virtual screening, and SBDD in developing functionally selective compounds for the M_1_ mAChR. However, more work is needed to establish optimal doses that preserve efficacy yet produce minimal side effects and, more generally, to better understand how occupancy of the mAChRs translates to efficacy over time and dose. For example, at what stage of AD would an agonist (or PAM) provide the best symptom relief? While there are still many unanswered questions, a few critical features of successful mAChR PAMs do appear to be consistent: (1) low cooperativity (e.g., low α-values); (2) low intrinsic activity; (3) and most importantly, high functional selectivity, appear to be essential factors for mAChR PAMs (or agonists) that have meaningful efficacy and are devoid of unendurable AEs. 

## 3. Drug Design Targeting the mAChRs: Schizophrenia

Schizophrenia is a chronic, disabling brain disorder affecting ~1% of the population (hhs.gov accessed on 15 December 2021) with multiple symptoms, some of which overlap with AD. The term schizophrenia comes from the Greek for “splitting of the mind [121]”. The onset of these markedly variable symptoms typically occurs early in life (e.g., <30 years of age), between adolescence and young adulthood, and has been hypothesized to be instigated by factors that are not well understood [122]. Characteristic symptoms include the positive cluster, including hallucinations, delusional thoughts/beliefs, and difficulty concentrating and speaking. Negative symptoms, which have not been well-targeted to date in the development of novel antipsychotics include anhedonia, dysfunctional social interactions, loss of interest in everyday life activities, and cognitive disruptions affecting several aspects of daily living (e.g., executive functions) [123,124]. It should be noted that while many AD symptoms are cognitive in nature (learning and memory), a large portion of AD patients (50–80%) also show psychotic and behavioral disturbances that correlate to poor social and functional outcomes similar to SZ [125].

Although the causes of the disease remain largely unknown, most of the research over the past few decades has targeted the dysregulation of the signaling by monoamines (e.g., dopamine and serotonin). The dopamine hypothesis of SZ (for review, see Carlson [126]) proposes that positive symptoms correlate with hyperdopaminergic activity in the striatal and mesolimbic pathways. In contrast, negative symptoms are related to hypodopaminergic activity in the medial prefrontal and mesocortical pathways [127,128,129,130]. This has been confirmed through positron emission tomography (PET), which showed that SZ patients had increased synaptic dopamine levels, released higher levels of dopamine in response to amphetamines, and an increased basal level of dopamine synthesis [131]. Current antipsychotics effectively treat positive symptoms but do not address the flattened affect, social withdrawal, working memory deficits, and cognitive flexibility [132]. Unfortunately, prescription treatments are commonly discontinued due to the extrapyramidal symptoms that patients experience (dystoria, akathisia, parkinsonism, bradykinesia) from first-generation treatments or metabolic side effects (e.g., weight gain, type II diabetes, hyperlipidosis) from second-generation atypical antipsychotics [133,134]. Therefore, given the detrimental consequences of the negative and cognitive symptoms of the disease, therapeutic strategies that move beyond D_2_ antagonism are needed. 

While the dopamine hypothesis of SZ served to introduce several generations of antipsychotic medications, it has long been recognized that dopamine hyperactivity alone cannot account for the wide range of positive, negative, and cognitive symptoms experienced by patients [135,136]. For example, in recent years, intriguing hypotheses have been presented by several groups concerning the role of post-synaptic density (PSD) proteins and their relationship to glutamatergic signaling in SZ. These proteins are a part of a specialized complex located at excitatory glutamatergic post-synaptic terminals, that contain NMDARs, membrane channels, scaffolding and signaling proteins, GTPases, kinases, and regulator proteins, among others that are implicated in synaptic plasticity [137]. Within the post-synaptic density region, tuning of downstream glutamatergic neurotransmission occurs, which allows for cross-talk between other neurotransmitter signaling pathways [138]. As multiple postsynaptic signaling pathways within this region of the neuron are involved in the transduction of glutamatergic and other postsynaptic signals, defects in these proteins (or disruption of protein clusters) may be implicated in SZ and other developmental disorders (e.g., autism, mood disorders—for a recent review see de Bartolomeis et al. [139]). 

Additionally, the psychotomimetic effects of NMDA/mAChR receptor antagonists, growing evidence that inhibitory GABA signaling is dysregulated in SZ patients (particularly in the cortex) [140,141,142,143,144,145,146], and the recognition that all three signaling pathways can be modulated through the activation of the cholinergic system has renewed interest in designing modulators of the mAChRs and nAChRs (for review of nAChRs see Taly et al. [147]). Since AChEIs have demonstrated therapeutic efficacy in mediating cognitive deficits in AD patients, it has been suggested that AChEIs might be effective as a supplementary medication in SZ patients [148,149]. However, clinical trial results have been disappointing due to dose-limiting cholinergic side-effects (primarily mediated by stimulation of the M_2_ and M_3_ mAChRs) of traditional non-selective cholinergic therapies [150,151]. 

Current strategies for targeting the mAChRs in the treatment of SZ are based on some key observations: 1) M_1_ and M_4_ KO mice exhibit a dopamine hypersensitivity phenotype [7], i.e., they are hyperactive compared to wild-type, and this effect is greater in M_1_ than M_4_ KO mice) [7,152,153,154]. Gerber et al. [155] reported that M_1_ KO mice displayed a significant increase in extracellular dopamine concentrations in the striatum, resulting in increased locomotor activity. These mAChRs are also usually responsible for regulating excitatory cholinergic flow to midbrain dopaminergic neurons [156]. This evaluation of hyperactivity derives from: (1) microdialysis studies showing that psychostimulants (*d*-amphetamine and phencyclidine) enhance dopamine efflux within the nucleus accumbens (nAcc) in the absence of M_4_ mAChRs; (2) postmortem studies (brain-tissue) indicated a reduction in expression levels of both M_1_ and M_4_ mAChRs caudate, putamen, hippocampus, cingulate cortex and the prefrontal cortex in SZ patients [135,157]; and (3) the M_4_ receptor is highly coexpressed with the D_1_ receptor [158]. Thus, when an agonist (e.g., oxotremorine) is administered to these neurons, a decrease in dopamine-stimulated cAMP is observed due to cross-talk between the D_1_/M_4_ receptors. cAMP signaling is activated (by dopamine) via the Gα_s_-coupled D_1_ dopamine receptor, and ACh inhibits cAMP production via the Gα_i/o_-coupled M_4_ receptor in a complex push-pull mechanism [159]. These results suggest that both M_1_ and M_4_ mAChRs play a role in controlling dopamine signaling in key regions of the brain and can be linked to increased locomotor activity in mice.

As drug development efforts to identify new mAChR subtype-selective ligands (specifically for M_1_ and M_4_) became possible, a number of studies using postmortem tissue (for review, see Hopper et al. [160]) reported decreased binding densities with [^3^H]-pirenzepine ([^3^H]PRZ), a selective M_1_/M_4_ antagonist, in the cortex, hippocampus, and striatum in SZ patients [161,162,163,164]. The link between decreased receptor densities was further strengthened by neuroimaging data showing a decrease in mAChR binding potential in patients not taking antipsychotic medication at the time of imaging [165]. Importantly, the decreased cortical binding levels that were evident in SZ patients were not evident in patients diagnosed with MDD, bipolar disorder (BPD), AD, or Parkinson’s disease [166]. Finally, the use of receptor-specific antibodies in immunopharmacological experiments and the quantification of mRNA levels in molecular biology experiments provided additional evidence of decreased levels of both M_1_ receptors in the dorsolateral prefrontal cortex and M_4_ receptors in the hippocampus in SZ patients [167,168]. In conjunction with subsequent reports of SZ-specific CHRM1 (M_1_) and CHRM4 (M_4_) polymorphisms, the above findings suggest that the disease and its distinguishing phenotype might be linked to a decrease in M_1_/M_4_ function [169,170]. This possibility has driven the strategy of stimulating the M_1_/M_4_ receptors as a means of alleviating the negative symptoms of SZ. 

It should be mentioned that ~25% of SZ patients have shown a reduction of up to 75% of normal M_1_ receptor expression in the prefrontal cortex (Brodmann area 9, BA9) when compared to non-SZ controls [156,171,172,173]. SZ patients with this reduced expression of M_1_ mAChRs have been characterized as having “muscarinic receptor-deficit syndrome” (MRDS). Dean et al. [156] and Hopper et al. [174] identified this phenomenon in binding experiments with [^3^H]-N-methyl-scopolamine ([^3^H]-NMS) and BQCA, showing that the ability of BQCA to modulate the displacement of [^3^H]-NMS by acetylcholine was reduced in MRDS tissues, which is similar to the decrease in cortical M_1_ receptors in SZ MRDS patients. This finding is clinically relevant and suggests that treating such SZ patients with M_1_ selective ligands may not be therapeutically advantageous, as the desired effect could be absent due to low CHRM1 levels. While this idea is speculative, it illustrates the types of challenges to the successful development of novel therapeutics for the treatment of the varied symptomology—positive, negative, and cognitive—that must be faced in undertaking such efforts. 

### 3.1. Orthosteric Agonists

Drugs that activate the M_4_ mAChR, albeit without appreciable subtype selectivity, are currently being utilized in the treatment of SZ. This strategy was introduced with the evaluation of Xanomeline, a partial agonist at the M_1_/M_4_ subtypes of mAChRs, as a treatment medication in SZ patients [175]. As noted above, dose-limiting side effects of Xanomeline (GI distress, likely due to agonism at peripheral mAChRs) limited its clinical effectiveness for the treatment of AD patients. Despite such setbacks, the success of Xanomeline in reducing behavioral disturbances and its positive effects on cognition warranted a small follow-up phase II trial [44]. Following two weeks of treatment, patients (who discontinued antipsychotic medication during the trial) showed significant improvements in the Brief Psychiatric Rating Scale (BPRS), Positive and Negative Syndrome Scale (PANSS), and Clinical Global Impression Scale, when compared to the placebo-controlled group. Again, as with the AD study [43], the side effects halted further clinical development. Renewed interest in Xanomeline (Figure 1) emerged from Karana Pharmaceuticals advancement of KarXT, a combination of Xanomeline and a peripherally restricted mAChR antagonist trospium chloride (to mitigate the adverse peripheral side effects of Xanomeline), which recently completed Phase II trials for SZ (NCT03697252) with promising results in the Positive and Negative Syndrome Scale (PANSS). The dose selected for the study (125 mg Xanomeline/30 mg trospium chloride) still produced cholinergic and anticholinergic side-effects [176,177]; nevertheless, Phase 3 clinical trials are now underway to evaluate efficacy, safety, and long-term safety and tolerability of KarXT in adult SZ patients (NCT04659161; NCT04738123; NCT04659174; NCT04820309). Large-scale clinical studies are still needed to fully determine the efficacy of KarXT; notwithstanding this caveat, the development of Xanomeline, though challenging, represents a novel therapeutic strategy for the treatment of SZ. 

Although little attention has been given to further advancing orthosteric M_4_ agonists, the drug development division of Sumitomo Dainippon Pharma in Japan published recent progress in developing dual M_1_/M_4_ agonists (akin to Xanomeline) via the identification of scaffolds through HTS that were then coupled with known pharmacophores of M_4_ PAMs and hybridized to discover lead compounds. Through their rather extensive SAR efforts, novel, highly selective M_1_/M_4_ agonists based on N-substituted oxindole (blue, 7, Figure 5) [178], dihydroquinazolinone (green, 8, Figure 5) [179], and 7-azaindoline (red, 9, Figure 5) [180] were developed and evaluated in vitro. Compound 7 partially activated M_1_ (EC_50_ = 12 nM) and M_4_ (EC_50_ = 29 nM) while showing negligible off-target binding and potent CNS penetration. Dihydroquinazolinone (8) was shown to be a high-efficacy M_1_ (81%) agonist but a partial agonist at M_4_ (49%, 0.3 μM) with good brain penetration and reversed methamphetamine AHL in rats (ED_50_ = 3.0 mg/kg, sc). Further SAR exploration led to the introduction of the methanesulfonyl group into the azaindoline scaffold, which decreased M_1_ agonistic activity with no loss of activity at M_4_. A simple addition of a methyl group on the N-carbethoxypiperidine led to compound 9, which displayed high bioavailability (49%, 1 mg/kg, p.o.) in rats, good brain penetration (brain/plasma ratio: 0.9), and an EC_50_ of 13 nM (M_4_ IA = 81% at 3μM) [180]. No further data have been reported on this or on related compounds; however, these studies provided some insights into key structural elements that can confer M_4_ subtype specificity (e.g., a basic piperidine core with a terminal ethyl carbamate functional group [purple, Figure 5]). 

Heptares Therapeutics (in partnership with Allergan) presented HTL0016878 (structure not known), a first-in-class highly selective M_4_ agonist that entered clinical trials in 2017 (NCT03244228; NCT04849286) for the treatment of neurobehavioral symptoms of AD. Nothing further is known about the compound or its progression to further clinical trials. Yet, it remains interesting as the only orthosteric M_4_ mAChR drug candidate to advance out of preclinical development. 

Pfizer scientists, using carbamate isosteres (e.g., amide, urea, sulfonamide), also have recently sought to develop novel M_4_-selective orthosteric agonists. These efforts quickly encountered the challenge of retaining M_4_ activity, leading to a shift in focus to exploring heteroaromatics. Pyrazine (orange, Figure 5) and 1,2,3-thiadiazoles (similar to Xanomeline) were identified as effective ethyl carbamate replacements, resulting in lead compound **10** that demonstrated ~29-fold selectivity for M_4_ [EC50 = 0.296 μM] over M_2_ [ > 8.83 μM] [181]. Due to a high conservation in the orthosteric site, recent efforts in the Pfizer program have shifted focus again, although to designing functionally selective allosteric modulators targeting the M_1_ and M_4_ receptors this time (see below). It is noteworthy that, despite some impressive preclinical data from a number of M_1_/M_4_ discovery programs, there has been little success in getting these lead compounds into clinical trials, except for Xanomeline (KarXT) and HTL-9936. 

### 3.2. M_4_-Positive Allosteric Modulators

Drug development efforts for the M_4_ mAChR have not been as plentiful as those for M_1,_ which is likely in part due to the high orthosteric homology between M_2_/M_4_ [10]. Lazareno et al. [76] reported that thiochrome (the oxidation product of thiamine, Figure 6) was found to increase the affinity of ACh 3–5 fold for inhibiting [^3^H]NMS binding to M_4_ receptors but did not affect ACh affinity at M_1_/M_3_/M_5_. Additionally, it decreased the direct binding of [^3^H]ACh at M_2_ receptors (0.1 mM), making it the first M_4_ PAM to be identified. Shortly after, Lilly’s campaign (launched in 1997) to identify an M_4_ PAM produced lead compound LY2033298, followed by Vanderbilt’s VU0010010 as first-generation M_4_ allosteric modulators based on the thieno[2,3-*b*]-pyridine core in 2008 (blue, Figure 6). 

Achieving selectivity at the orthosteric site proved to be a daunting task, as evidenced by unwanted cholinergic AEs in early clinical trials with Xanomeline, hence Lilly’s efforts focused on developing M_4_ PAMs. Lead compound LY2033298 displayed robust cooperativity to ACh and had suitable physicochemical properties for in vivo dosing [181]. It potentiated the behavioral effects of the nonselective mAChR agonist oxotremorine (OXO-M) in reversing apomorphine-induced disruption of pre-pulse inhibition and conditioned avoidance responding [182,183]. Nevertheless, it lacked utility for studies in rodents due to its low potency at rat M_4_ ([rM_4_], 5–6-fold reduction from human M_4_ [hM_4_] mAChRs) and displayed higher cooperativity for OXO m than the endogenous ligand, ACh [182]. The introduction of an N-methyl acetylpiperazine functional group generated a close relative to LY2033298, LY2119620, which proved to be unsuitable as a therapeutic due to lack of specificity (M_2_ cross-reactivity) [184]. 

Furthermore, VU0010010 was developed from a chemoinformatics and medicinal chemistry approach to identify a series of highly selective allosteric potentiators of rat M_4_. The compound demonstrated robust and potent (EC_50_ = 400 nM) activity at M_4_ and potentiated ACh’s response 47-fold while showing no activity at other mAChR receptors [185]. Despite showing structural similarity to LY2033298, it lacked the physicochemical properties (log P~4.5) necessary for in vivo dosing (i.p. doses were not centrally active). Nevertheless, the compound served as a valuable in vitro tool and was further optimized to produce VU0152099 and VU0152100 (Figure 6) [186]. Both compounds demonstrated high M_4_-subtype specificity with comparable EC_50_ values to VU0010010 (~400 nM) and were CNS penetrant after systemic administration. Importantly, both VU0152099 and VU0152100 demonstrated efficacy in reversing AHL in rats/WT mice (not M_4_ KO mice) [187,188] when administered alone, implying that there is a sufficient amount of endogenous ACh to mediate these behaviors for modulation with M_4_ PAMs. This result further supports the hypothesis that endogenous ACh plays a vital role in regulating dopaminergic control of motor function. 

These early scaffolds paved the way to newer generations, with the efforts of Vanderbilt at the forefront. Optimization of VU0152100 led to ML173 (Figure 6), which showed more than an order of magnitude greater potency for human (EC_50_ = 95 nM) versus rat (2.4 μm) M_4_ mAChRs [189,190], and VU0448088 [ML253] (hM_4_ EC_50_ = 56 nM, rM_4_ EC_50_ = 176 nM) [191]. Wood et al. [192] detailed the SAR of a 5-amino-thieno[2,3-c]pyridazine (orange, Figure 6) series that expanded on the 3-amino-thieno[2,3-*b*]-pyridine core to produce VU0467154, which displayed a favorable DMPK profile but showed preferential potency (~35 ×) for rM_4_ versus hM_4_, preventing clinical use. 

Multiple optimization efforts on the benzene ring, which was deemed a metabolic hot spot due to CYP-mediated oxidative demethylation, afforded VU0467485/AZ13713945 (in collaboration with AstraZeneca). This was one of the first compounds that showed comparable activity at both hM_4_ (EC_50_ = 78.8 nM) and rM_4_ (EC_50_ = 26.6 nM) and displayed no significant species differences in potency (e.g., dog, monkey). Additionally, it showed robust in vivo efficacy preclinical models of SZ and a favorable DMPK profile in rats [193]. Due to high projected therapeutic doses (>450 mg, TID), low CNS penetration, and solubility issues, the compound failed to advance further. Other core scaffolds such as a benzothiazole (e.g., VU0409524, not shown) [194], thieno[2,3-d]pyrimidine (VU6002703, not shown) [195], 6-fluoroquinzoline (VU6003130, not shown) [195], and tricyclic triazolo- and imidazopyridine lactams (VU6005877, not shown) [196] have been reported; however, none have reached clinical evaluation to date. 

Most recently, Schubert et al. (Merck) [197] conducted an M_4_ PAM fluorescent imaging plate reader high-throughput screen (FLIPR HTS) of an extensive library of compounds to identify 2,3-substituted pyridine 11 (Figure 7). This was one of the few hits that displayed similar rat and human M_4_ PAM activity and had excellent subtype selectivity (M_1–3_ > 30 μM). Its limiting factor was the rate of efflux by P-gp but it served as a model scaffold which the group sought to optimize. After extensive optimization of the aromatic group (red circle), substitution on the core pyridine ring (green circle), and hydrocarbon tail group (blue circle), the group arrived at 12. This lead compound displayed a relatively similar activity in humans (IC_50_ = 17 nM) and rats (IC_50_ = 29 nM) and was over 100× selective for M_4_ compared to all other subtypes. Additionally, it showed high brain permeability and was shown to reduce AHL in rats with relatively mild cholinergic side-effects (even at much higher doses needed for reduction of AHL). Compound 12 was instrumental in developing a highly selective M_4_ PET-ligand ([^11^C]MK-6884] for mapping M_4_ receptors [198] in AD and ultimately led to MK-4710, which was first disclosed in the Fall 2020 ACS meeting [199]. However, the SAR needed to arrive at this lead structure and plans for clinical development have not yet been revealed. 

Despite several setbacks, there is currently a novel M_4_ PAM, CVL-231 (structure unknown), in clinical development from Cerevel Therapeutics (in partnership with Pfizer). It is now in Phase 1b clinical trials that will evaluate the safety, tolerability, and PK/PD of multiple ascending doses of the drug (NCT04136873). The collective data from these above-listed compounds suggest that M_1_ and M_4_ subtype-selective PAMs could be valuable tools for elucidating these receptors’ exact role in SZ and may represent a novel therapeutic approach for managing symptoms. However, additional studies are needed to characterize the pharmacological properties that translate to clinical efficacy versus adverse cholinergic effects.

It is important to note that although none of the described M_1_ PAMs (in previous sections) have undergone clinical trials for SZ, evidence supports their therapeutic applications in managing the negative symptoms of the disease. Previous work has determined that the M_1_ mAChR is coupled to the N-methyl-*d*-aspartate subtype of the glutamate receptor (NMDAR), and M_1_ activation potentiates NMDAR signaling in both the cortex and hippocampus, both of which are associated with learning and memory [200,201,202]. Additionally, as noted above, M_1_ KO mice also show hyperactivity relative to wild-type controls, suggesting that both receptors likely play a role in controlling locomotor activity [7,152,153,154].

Furthermore, VU0453595 (Figure 4) was found to be able to rescue cognitive disruptions, long-term depression (LTD), and restore social interactions in mice treated repeatedly with PCP (a mouse model of SZ) [203]. Studies with VU6005246 (difluoro-1H-indole core, green, Figure 5) showed that it dose-dependently reduced cognitive disruptions and hyperlocomotion in GluN1 knockdown mice (a genetic mouse model of NMDA hypofunction), and reversed performance impairments in both the NORT and fear conditioning tasks [204]. These results indicate that these cognitive effects are at least in part mediated by M_1_.

Spalding et al. [87] and Sur et al. [205] demonstrated N-desmethylclozapine (Figure 4, N-demethylation shown in red), a metabolite of the antipsychotic clozapine, was also a functionally selective M_1_ agonist that potentially acted through an allosteric mechanism. It was shown to dose-dependently potentiate NMDA receptor currents in CA1 pyramidal cells (widely accepted to be mediated by activation of the mAChRs), perhaps contributing to its clinical utility for schizophrenic patients. Multiple other M_1_ PAMs such as BQCA [93], PQCA [206,207], PF-06764427 [208], and PF-06767832 [209], among others, have all contributed to the proposal that activation of M_1_ receptors could have an essential role in medial prefrontal cortex (mPFC)-dependent cognitive functions and modulation of the mesocortical dopamine system [210]. However, more work is needed to determine the effects of the M_1_ mAChR on dopamine release. 

## 4. Drug Design Targeting the mAChRs: Major Depressive Disorder 

Janowsky and colleagues initially proposed the cholinergic hypothesis of depression in the early 1970s, postulating that hyperactivity of the central cholinergic system plays a key role in the pathophysiology of depression [211]. In short, the hypothesis states that mania is likely due to an imbalance of adrenergic activity, while depression is the result of high cholinergic activity [212]. Early evidence was found by demonstrating that direct-acting mAChR agonists (e.g., arecoline) and indirect-acting cholinergic agonists (e.g., AChEI’s such as physostigmine) could produce rapid and profound effects on mood in patients with mood disorders. The enhancement of cholinergic activity with physostigmine (via blocking the degradation of Ach) induced symptoms of depression in healthy human subjects and produced a shift in mood state from euphoria to depression in manic patients with BPD [213,214,215,216]. Further support for these early findings came when the administration of arecoline and oxetremorine (direct-acting agonists) was shown to worsen the mood state in MDD [217,218] while scopolamine (a nonselective mAChR antagonist) blocked these effects [219]. 

The mAChRs were explicitly implicated in these effects by evidence showing that polysomnographic responses to selective mAChR agonists were much greater in depressed patients than in control samples, suggesting that muscarinic receptor hypersensitivity existed in depressed individuals. The cholinergic system in the brain stem plays a significant role in activating the rapid eye movement (REM) and the non-REM sleep cycle. When cholinergic agonists such as RS86, arecoline, and physostigmine were given to patients with MDD, the sensitivity of the REM sleep response increased [220]. In contrast, scopolamine, a mAChR antagonist, decreased REM sleep and increased sleep latency in patients with depression [221,222]. Additionally, it was shown by various groups [223,224,225] that variations in *CHRM2* (gene encoding for the M_2_ mAChR) were associated with a higher frequency and severity of unipolar depression and with abnormal reductions in M_2_ receptor binding in bipolar depression. 

Sex-related effects were also present in mood disorders related to abnormal cholinergic receptor function. In premenopausal females with MDD, there was a higher frequency of heightened cholinergic sensitivity [226,227]. Comings et al. [224] found that genetic variants in the CHRM2 gene (A/T 1890 polymorphism in 3′ UTR of CHRM2 gene, homozygotes) were directly associated with MDD in female subjects. Other studies by Luo et al. [228] on the CHRM2 gene revealed modest associations between variation in gene sequence and populations of European-Americans/African Americans with affective disorders. Furthermore, a combination of SNPs, commonly seen (~40%) in patients with alcohol dependence and MDD, at the 5′ end of the CHMR2 gene was found to be associated with these two diseases based on the collaborative study on the genetics of alcoholism [228] although other clinical studies have produced conflicting results [229,230]. While there are still many unanswered questions regarding the genetic associations underlying depression, a few general hypotheses have been suggested: (1) M_1_ receptors provide a reliable target to improve cognitive deficits (as described by Merck and others) [231], (2) the M_2_ mAChR has been associated with MDD based on single nucleotide polymorphism (SNP) studies (above) and KO mice, and (3) Gibbons et al. [232] reported a decreased binding of M_2_/M_4_ mAChRs in the dorsolateral prefrontal cortex of depressed subjects.

The cholinergic hypothesis was further supported by animal studies investigating the psychopharmacological effects of antidepressants on the CNS. Flinders Sensitive Line rats (FSL), bred selectively for increased sensitivity of mAChRs, are considered to model certain aspects of depression, including loss of appetite, increases in REM sleep (lethargy), reductions in self-stimulation, learning difficulties, and decreased mobility in the forced swim test (FST) in response to agents that stimulate cholinergic function. However, it does not mimic all the biochemical aspects of depression [233,234]. Although we will focus on drugs developed for the mAChRs in this discussion, the nAChRs deserve mention since they also play a vital role in cholinergic signaling (see Dulawa and Janowsky [235]).

There are several classes of drugs, such as monoamine oxidase inhibitors (MAOIs), tricyclic antidepressants (TCAs), selective serotonin reuptake inhibitors, and serotonin and noradrenaline reuptake inhibitors (SNRIs) that exist today for the treatment of depression. The drugs were designed based on the monoamine theory of depression that predicts the underlying pathophysiologic basis of depression, i.e., a depletion in serotonin, norepinephrine, and/or dopamine levels in the CNS [236,237]. Although differing in their selectivity for specific receptors or transporters, their primary function is to enhance either serotonergic or noradrenergic neurotransmission. It was concluded early on by El-Fakahany [238] (and more recently by Siafis and Papazisis [239]) that most, if not all, were antagonists of the mAChRs with the tricyclic antidepressants (e.g., amitriptyline, butriptyline, trimipramine, etc.) showing higher efficacy in MDD patients [240,241,242]. Janowsky et al. [211] and Davis et al. [243] postulated that the antimuscarinic properties (e.g., a mAChR blockade) were responsible for the mood-elevating effects (reviewed in Dagytė et al. [213]). However, these drugs have been shown to have many other actions associated with their affinity for histamine, dopamine, serotonin, adrenergic, and the mAChRs [237].

Little has been produced that is more effective than the original TCAs or MAOIs despite over 50 years of research in the field. While antidepressant drugs have been beneficial to many, according to the NIMH-funded STAR*D (Sequenced Treatment Alternatives to Relieve Depression) clinical trials, 47% of patients’ symptoms are not adequately treated within the first antidepressant regimen (usually an SSRI), and 30% after four different treatments [244,245,246]. Additionally, currently approved drugs must typically be administered 4–6 weeks before a therapeutic effect is achieved. This delayed action is not well understood and presents a significant challenge for medical professionals in managing MDD. This delay prolongs patient suffering, delays the implementation of potential new treatments, and increases the risk of disease-related health issues (including death by suicide). 

Furthermore, many available antidepressant medications have adverse side effects, such as anxiety, agitation, suicidal ideations, and declines in short-term memory and cognitive function [247,248]. Chronic use is accompanied by a cognitive disruption in 30% of subjects [249], uncomfortable symptoms following discontinuation [250,251], and the drugs have been recently found to increase the likelihood of developing dementia in old age [252,253,254]. Not surprisingly, the urgent need for novel antidepressants with an improved side-effect profile has been noted by multiple investigators [255,256]. Later generations of antidepressants (e.g., bupropion, citalopram, duloxetine, fluoxetine, sertraline, etc.) have improved selectivity for their respective transporters (hence more favorable side effect profile), yet still have considerable limitations in treating the condition [241,257]. 

To properly treat patients suffering from mood disorders, the urgent need persists for rapid antidepressant therapies that avoid the delayed onset of classical antidepressants. This is important not only for those suffering from MDD but also for a small subset of patients whose treatment response is minimal or absent altogether [258,259,260]. Treatment-resistant depression (TRD) severely limits treatment options for clinicians, despite accounting for about 30% of all patients suffering from depression. Some reassurance came from replicating earlier findings by Berman et al. [260], confirming NMDA receptor channel blocker ketamine produced rapid improvement in mood in TRD patients [261]. The results were confirmed again in a small clinical trial by Zarate et al. [262] and confirm the NMDA blockade hypothesis initially proposed by Trullas and Skolnick [263] about three decades ago. In March 2019, intranasal (S)-ketamine (esketamine or Spravato) received FDA approval to be administered as a supplemental treatment for adults with TRD. The approval was based upon the efficacy of esketamine in short-term clinical trials [264] and longer-term maintenance investigations (focusing on individualization of treatment) [265](NCT02497287/NCT02782104). It is worthy of mention that ketamine has been shown to cause dizziness, vertigo, headache, increases in blood pressure, and to produce dissociative effects [266,267]. In addition to these safety concerns, ongoing studies are underway to assess the durability of the antidepressant effects of ketamine, with maintenance dosing being at the forefront [268,269].

Mechanisms of antidepressants have been a significant focus of current and past research to develop more effective and faster-acting drugs. In studies to assess whether a reduction in mAChR function would alleviate depressive symptoms, clinical data showed that the nonselective mAChR antagonist scopolamine (Figure 8) had antidepressant effects in patients with either MDD or BPD [270]. No participants in the trial dropped out due to adverse side effects, likely attributed to the low dose (4 μg/kg). However, scopolamine and other known anticholinergic drugs have side effects, especially with long-term use. These include double vision, increased heart rate, confusion, disorientation, euphoria/dysphoria, memory problems, mental confusion, or other CNS effects that resemble delirium [271,272,273]. One of the downsides of utilizing scopolamine as an antidepressant is cognitive impairment, especially at higher doses (e.g., ≥8 μg/kg). Since MDD is already associated with cognitive deficits [274,275], this would produce an undesired additive effect that would only exacerbate the problem. In summary, the cholinergic hypothesis of depression can be based on the following findings: (1) current antidepressants are high-affinity antagonists at the mAChRs; (2) cholinergic stimulation leads to symptoms of depression; (3) scopolamine produces relatively rapid and robust antidepressant effects even in those patients that were refractory to previous treatments; (4) the mechanism of scopolamine in part resembles that of ketamine (discussed below).

Scopolamine’s antidepressant effects mechanism is still not well understood, particularly its persistence of antidepressant action, due to its relatively short clearance time from plasma (t_1/2_ = 2–4 h)—although it has been shown to have a long residence time in tissues. This result suggests a mechanism other than the direct blockade of the mAChR receptors. The persistent antidepressant response that lasted long after the anticholinergic side effects (reported to have dissipated within 5 h in the clinical study) lends to the hypothesis that the drug may alter synaptic plasticity or gene expression through a variety of direct/indirect mechanisms [276]. 

To better understand the mechanism of scopolamine, Voleti et al. [277] examined the role between the mammalian target of rapamycin complex 1 (mTORC1) and synaptogenesis in the PFC, which has also been implicated in the rapid actions of NMDA antagonists. They discovered that a single, low dose (25 µg/kg) significantly induced phosphor-mTOR, phosphor-Akt, and phosphor-S6K 1 hr post-treatment (Western blot). Similarly, telenzepine (3 mg/kg, known to have limited selectivity for the M_1_ mAChR) showed increased expression of the same proteins. In congruence from earlier studies of ketamine [278], scopolamine administration rapidly increased the number of spines within the distal segments of layer V neurons in the PFC and produced a small but significant increase in spine head diameter are consistent with increased synaptic function [276]. As expected, scopolamine (25 μg/kg) showed antidepressant activity in the FST but was blocked upon pretreatment with the selective mTORC1 inhibitor rapamycin, similar to ketamine. The group also demonstrated that pretreatment with the known α-amino-3-hydroxy-5-methyl-4-isoxazolepropionic acid (AMPA) receptor antagonist NBQX blocked the antidepressant actions of scopolamine in the FST, confirming a requirement for glutamate-AMPA activity.

In a follow-up study by Navarria et al. [279], scopolamine microinfusions directly to the infralimbic (IL) and prelimbic (PrL) subregions of the mPFC produced responses in the FST. However, neuronal silencing of these regions blocked the effects. Additionally, systemic administration of VU0255035, a selective M_1_ antagonist, displayed an antidepressant response and increased mTORC1 signaling in the PFC. Their results also demonstrated that a single dose of either scopolamine or VU0255035 blocked the anhedonic response caused by chronic unpredictable stress (CUS). These results indicate that the mPFC is a vital mediator of scopolamine’s behavioral actions and points to the M_1_ receptor as a therapeutic target for developing novel and selective rapid-acting antidepressants. In further support of these above results, Witkin et al. [280] utilized both wild-type and KO mice to identify the subtypes responsible for the antidepressant action of scopolamine. M_1_ and M_2_ KO mice showed a diminished response to scopolamine (but not imipramine) in the FST, indicating that both of these receptors likely play a role in scopolamine’s antidepressant effects.

Wohleb et al. [281] also demonstrated the role of M_1_ mAChRs in mediating scopolamine’s antidepressant effects in rodents by subjecting the animals to chronic mild stress (a preclinical model of depression). By using a viral-mediated knockdown of M_1_ mAChRs specifically in GABAergic neurons, but not glutaminergic neurons, in the mPFC, scopolamine’s antidepressant effects are significantly diminished. Therefore, it can be concluded that blocking these receptors produces a short-term burst of glutamate that results in neuroplasticity effects at the glutamatergic synapse [281,282].

As noted previously, the selective M_1_ agonism (through selective orthosteric agonists or PAMs) has long been accepted as a method to enhance cognition, and consequently, M_1_ antagonism has been viewed as detrimental. However, scopolamine produced cognitive deficits in both wild-type and M_1_ KO, which could infer that antagonism of M_1_ may not be a cause for concern [271]. This, however, has yet to be tested experimentally. While the M_2_ receptor has been implicated in the pathophysiology of mood disorders and depression (detailed previously), its utility as a target could be compromised by its high prevalence in cardiac tissue [283]. 

There is less convincing evidence to support the targeting of M_3_, M_4_, and M_5_ receptors. M_3_ is enriched in the GI tract and bladder tissues but shows low levels of expression in the hippocampus and cortex [38,284,285]. M_4_ has a similar expression level to M_1_ in specific brain regions (hippocampus, cerebral cortex, striatum). In the striatum where M_4_ is most highly expressed with M_1_, dense patches of receptor expression correspond to postsynaptic sites on medium spiny neurons, which modulates GABA release. More significantly, M_4_ (like M_2_) is known to be an autoreceptor that inhibits the release of ACh; therefore, antagonizing this receptor (as with M_2_) would serve to enhance cholinergic signaling [162,163,164,165,286] and likely would not produce an effective antidepressant. Finally, M_5_ is expressed exclusively (and only in particular regions) in the CNS, albeit at low levels. These regions include the VTA projections to the nAcc, which modulates dopamine release, the hippocampus, substantia nigra, and peripheral and cerebral blood vessels [287,288,289]. As VTA dopamine release has been related to the mediation of stress responses, it could potentially mitigate depressive symptoms (via mediation of anhedonia-like behaviors) [271,290]. 

Aside from studies on scopolamine, there has been little published on muscarinic antagonists as potential antidepressants. In the past year, Johnson et al. [291] reported the development of two potential lead compounds (L-687306 and CJ2100, Figure 8) targeting the orthosteric site acting as rapid-acting antidepressants that lack cognitive deficits. Additionally, L-687306 was synthesized and evaluated as a possible treatment of AD in the early 1990s by Merck [292,293,294] to identify a more selective mAChR agonist with reduced efficacy (hence reduced cholinergic side-effect profile), and CJ2100 is a cyclopropyl oxadiazole derivative of arecoline. These compounds were compared with scopolamine in vitro and across several in vivo assays that reflected autonomic and central activity. As shown in Figure 9, each of these compounds displayed similar effects in the FST, decreasing immobility and increasing either swimming or climbing (suggesting antimuscarinic-mediated antidepressant activity). Additionally, scopolamine and CJ2100 reduced immobility in the FST without significantly altering locomotor activity at the same doses, leading the authors to conclude that the effects in the FST were independent of general activity stimulating properties. 

As a proof of principle to show the possibility of designing a muscarinic antagonist without cognitive deficits, the group also evaluated each compound in cognitive assays of attention (Psychomotor Vigilance Task) and memory (Titrating Delay Matching to Position Task). Both L-687,306 and CJ2100 showed no detrimental effects in either assay (up to 10 mg/kg) as opposed to scopolamine, which produced dose-dependent impairments in both tasks. Based on their results, it is possible to design functionally selective drugs that target the orthosteric site of the mAChRs. Additionally, it proves the utility of drug design methods targeting these receptors as a novel way to treat depression.

In summary, scopolamine has produced robust antidepressant effects in animal models of depression and has demonstrated efficacy in multiple clinical trials. The current findings thus far most clearly support that the antagonism of M_1_ should potentially provide a novel way to treat depression. However, emerging data also implicates the M_5_ receptor in modulating dopamine release in the midbrain, and as such, it would be appropriate to study whether antagonism at M_5_ could potentially mitigate anhedonic-like phenotypes in stress models (i.e., Nunes et al.) [295]. 

## 5. Conclusions

Despite advances in GPCR crystallography in solving the structures of the M_1_–M_4_ receptors, targeting the mAChRs in drug design remains an elusive task. Binding simulations and the vast array of crystal structures currently available have allowed drug discovery efforts to be expedited from traditional high throughput screening (HTS) to virtual screening or fragment-based drug design. Both latter methods require fewer resources and serve as a template for selecting potential lead molecules to be evaluated via other methods. These efforts have identified multiple orthosteric and highly selective allosteric modulators of the mAChRs to treat CNS indications, some of which have progressed to clinical trials (summarized above). 

Much remains to be learned about allosteric modulation of the mAChRs due to the complex interactions between the receptor, orthosteric ligand, and allosteric modulator. Alteration of any of these could lead to different cooperativity and subsequently skew the drug’s pharmacological profile interpretation. Additionally, unlike the orthosteric site that can be narrowed to specific key residues responsible for high-affinity binding (or functional selectivity), single residues cannot be identified that determine the type and strength of cooperativity. This is likely due to many domains playing a crucial role in the α-values. 

While allosteric modulation has primarily been the focus of recent efforts, it should not preclude the design of functionally selective orthosteric ligands, as demonstrated by the development of the highly selective M_1_/M_4_ ligands for AD/SZ and also functionally selective mAChR antagonists as a novel treatment for depression. A great amount of effort has been dedicated to targeting the mAChRs for treating CNS disorders, and the field continues to remain largely open for discoveries.

## Figures and Tables

**Figure 1 biomedicines-10-00398-f001:**
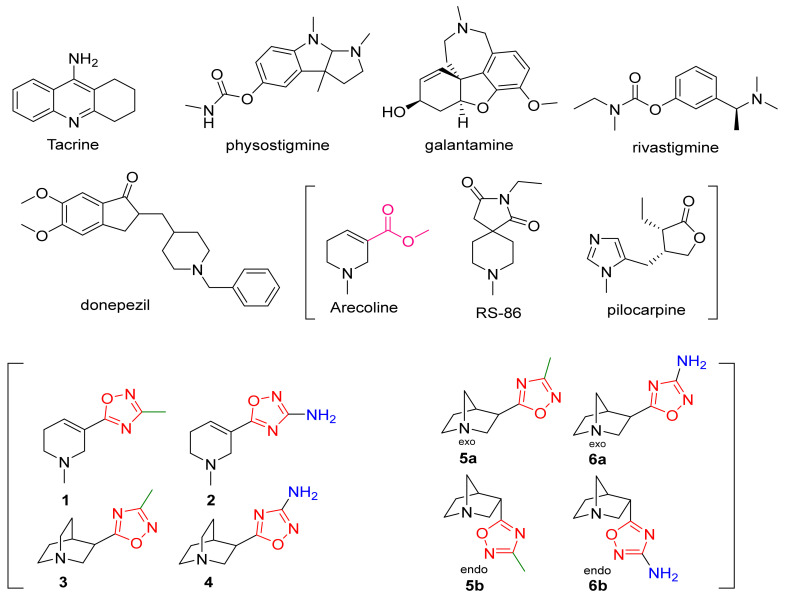
AChEs and direct acting mAChR agonists (brackets).

**Figure 2 biomedicines-10-00398-f002:**
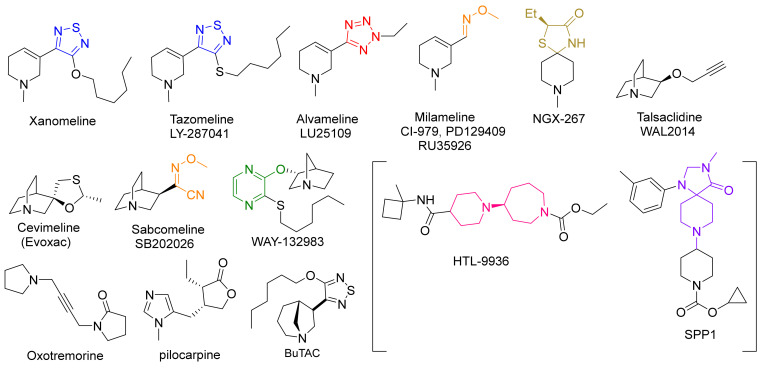
First generation of orthosteric mAChR agonists. HTL-9936 (Sosei Heptares, Phase 1 trials) and SPP1 (Eli Lilly, not yet in clinical development) represent recent efforts (brackets) targeting the mAChRs with functional selectivity for M_1_.

**Figure 3 biomedicines-10-00398-f003:**
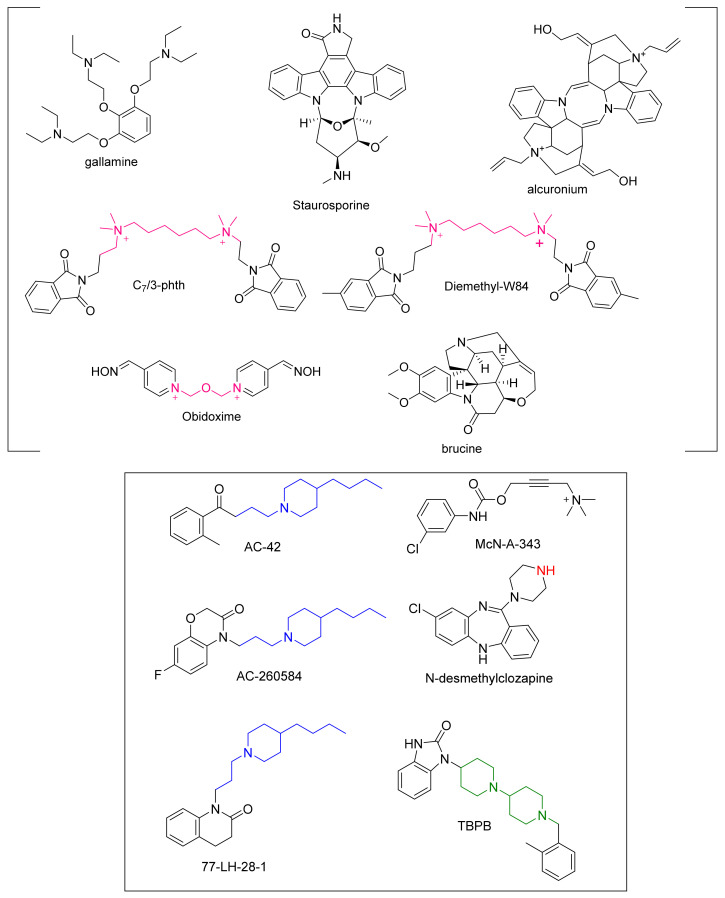
Neuromuscular blocking agents and bis-quaternary ammonium compounds (brackets). First generation of allosteric modulators (box).

**Figure 4 biomedicines-10-00398-f004:**
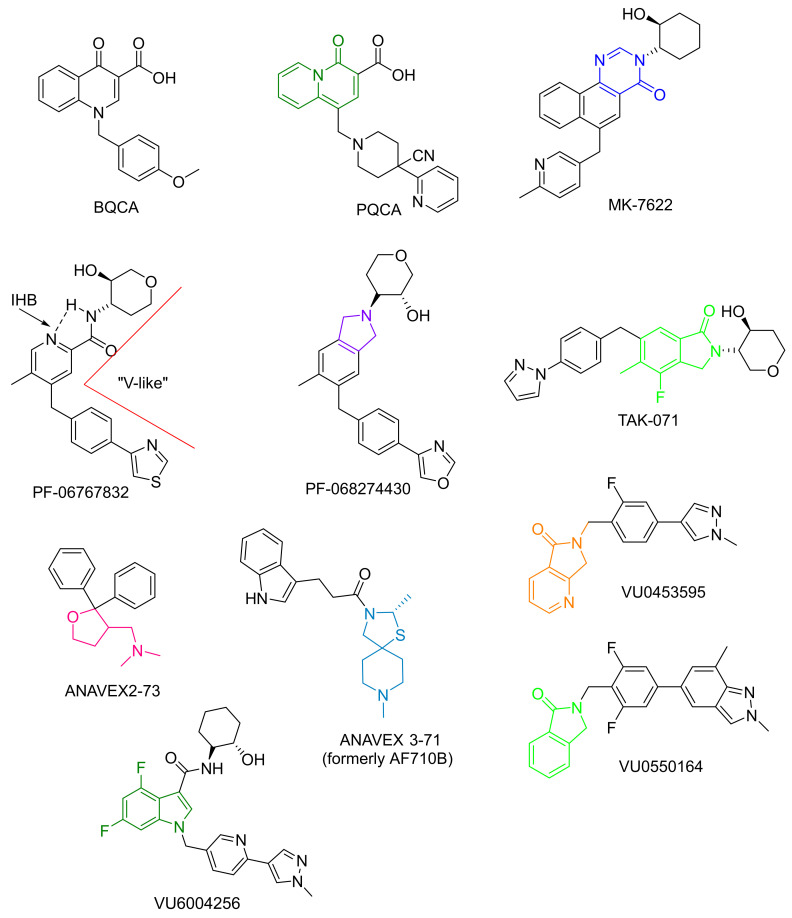
Recent designs of allosteric modulators (primarily) targeting the M_1_ receptor.

**Figure 5 biomedicines-10-00398-f005:**
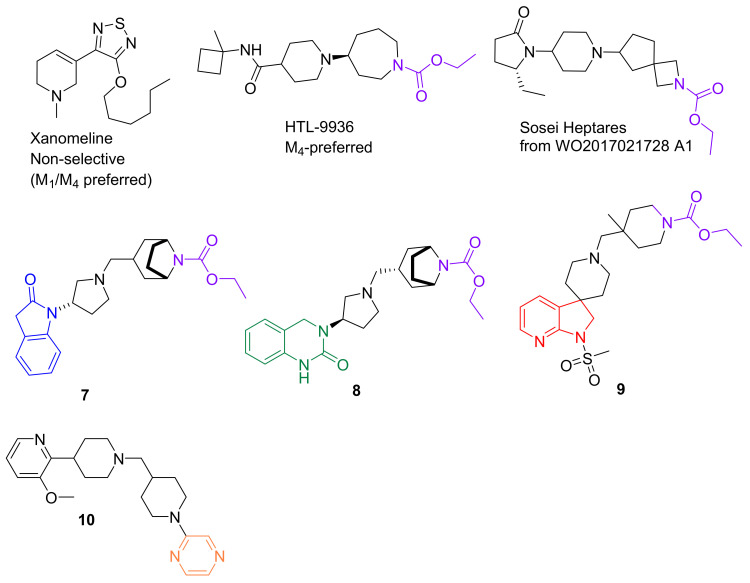
Recent examples of highly selective dual M1/M4 (or M4) agonists.

**Figure 6 biomedicines-10-00398-f006:**
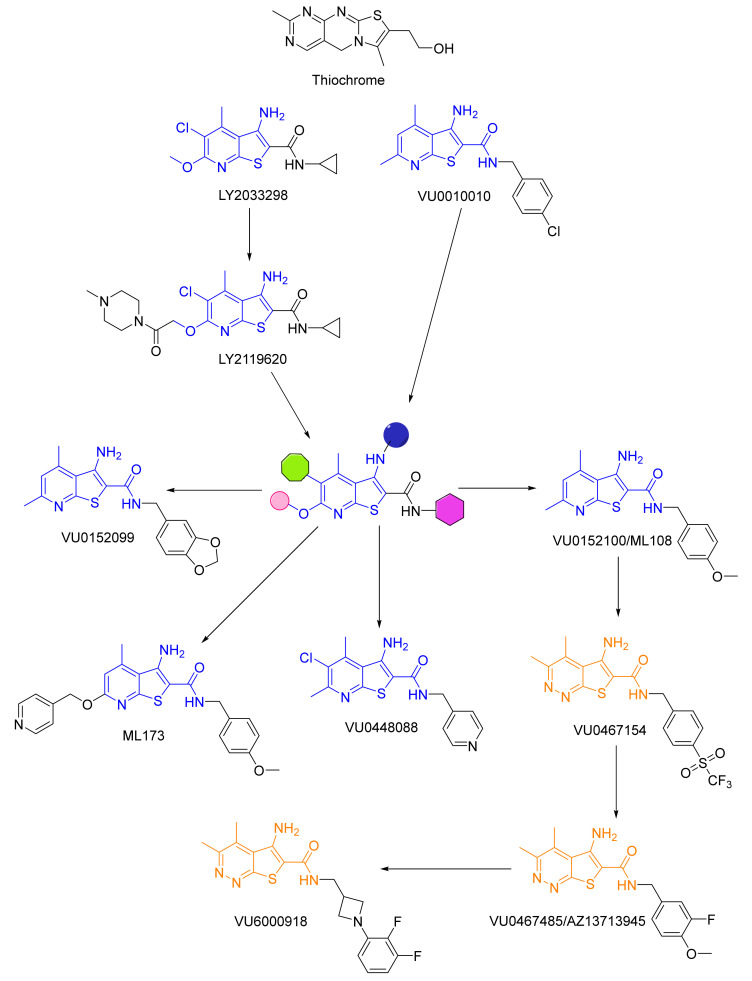
Optimization campaign to arrive at examples of highly selective dual M_1_/M_4_ (or M4) agonists.

**Figure 7 biomedicines-10-00398-f007:**
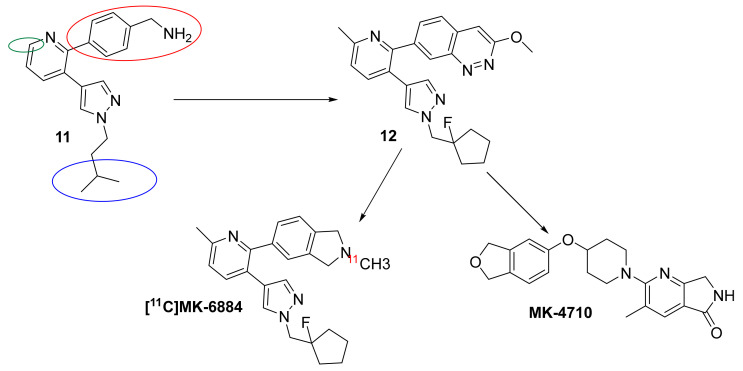
Optimized M_4_ selective PAMs recently disclosed by Merck.

**Figure 8 biomedicines-10-00398-f008:**
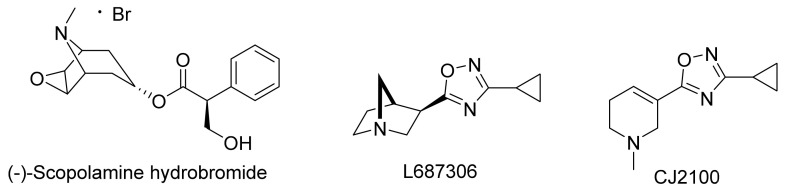
Examples of potential antidepressants targeting the mAChRs.

**Figure 9 biomedicines-10-00398-f009:**
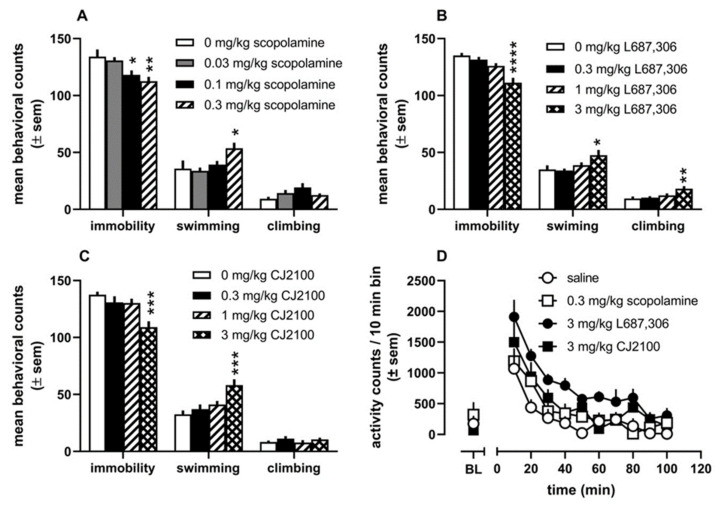
Effects of several doses of Scopolamine (**A**), L-687,306 (**B**), and CJ2100 (**C**) on counts of immobility, swimming, and climbing in the forced swim test. (**D**) Locomotor activity at doses that produced decreases in immobility in the FST. (**E**) Dose-related effects of scopolamine (squares), L-687,306 (diamonds), and CJ2100 (circles) on mean titrated duration (seconds) in the psychomotor vigilance task. (**F**) Dose-related effects of the compounds on the mean titrated delay (seconds) in the titrating delay matching-to-position task. * *p* < 0.05, ** *p* < 0.01, *** *p* < 0.001, **** *p* < 0.0001. Reproduced with permission from Johnson CR et al. [291].

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
