# Peer review of "Drug Design Targeting the Muscarinic Receptors and the Implications in Central Nervous System Disorders"

_biomedicines, 2022, doi:10.3390/biomedicines10020398_

Round 1

Reviewer 1 Report

This review by Johnson et al. discusses various medicinal chemistry approaches targeted at muscarinic receptor subtypes for the treatment of dementia, schizophrenia and depression. The manuscript is well written and illustrated with helpful structures in which typical core structures are emphasized by color. Discussion of the various approaches is adequate, and almost 300 references testify to the author´s diligence to achieve broad coverage of the literature.

  1. Overall, the approach of the authors is from a medchem perspective, but much insight is given into biological background, animal models and clinical outcomes. Nevertheless, I feel that the non-specialist reader may appreciate a short discussion (for instance, a text box) to illustrate some important points from pharmacology. For instance, the brain permeability of centrally acting drugs is of pivotal importance, and some information is given, but the parameters vary (what is “good” or “potent” brain permeation, lines 550-1). The brain to plasma ratio, for instance, leaves the reader uncertain whether brain homogenates are compared to plasma levels whereas the plasma – CSF ratio should be close if only the unbound part of the plasma level is used for comparison (CSF has very little protein). Also, the P-glycoprotein pops up occasionally but only with selected drug examples, and its relevance for brain permeation remains elusive for the non-specialist reader (some agonists are pretty big structures, see Fig. 5, and may be PGP substrates). There are other pharmacological phenomena which are poorly explained, e.g. spare receptors. Maybe a brief discussion of some of these points in a separate box would improve reader understanding.
  2. Another phenomenon that is not well discussed is the fact that acetylcholine, the physiologic agonist, is a permanent cation, and that it is difficult to model an orthosteric ligand that penetrates the blood-brain barrier and has agonistic action on receptors that are used to interacting with a cationic transmitter.
  3. Finally, from a pharmacological standpoint, continuous activation by agonists is clearly inferior to increasing the “cholinergic tone” e.g. by inhibiting AChE or using PAMs. This point is made by the authors but sort of hidden away among the lengthy drug discussion.

Further comments

  1. Line 424-425, please move this sentence to the end of the paragraph, after the discussion on schizophrenia. AD patients do not have cognitive dysfunctions below 30 years of age.
  2. The text in lines 898-904 is identical to lines 905-910, and the figure should be 9, not 8.

Minor comments

  1. Line 79, delete “was present”.
  2. On line 215-220, one wonders why a press release is cited but then the results were not “made public”: Do the authors mean that there was no publication with detailed information?
  3. Line 376 ff., it is somewhat surprising that compounds that were once developed for Alzheimer´s dementia (cholinergic hypofunction) were later repurposed to Parkinson´s disease (which is hypercholinergic in a way).
  4. Line 595: oxotremorine is non-selective in vitro, but more or less M2-specific in vivo (talk about spare receptors…).
  5. Line 812, do these effects of scopolamine (antidepressant and anti-cognitive) occur at the same doses?
  6. Line 822, scopolamine has a short clearance time from plasma but has a long residence in time in most tissues in the body, including the brain. Line 823, should be “blockade”, not “activation”.

Author Response

**Please note that the lines referenced in this response currently reflect our revised document that includes tracked changes. Therefore, the line numbers no longer match up to the ones initially mentioned in this review. **

1. Overall, the approach of the authors is from a medchem perspective, but much insight is given into biological background, animal models and clinical outcomes. Nevertheless, I feel that the non-specialist reader may appreciate a short discussion (for instance, a text box) to illustrate some important points from pharmacology. For instance, the brain permeability of centrally acting drugs is of pivotal importance, and some information is given, but the parameters vary (what is “good” or “potent” brain permeation, lines 550-1). The brain to plasma ratio, for instance, leaves the reader uncertain whether brain homogenates are compared to plasma levels whereas the plasma – CSF ratio should be close if only the unbound part of the plasma level is used for comparison (CSF has very little protein). Also, the P-glycoprotein pops up occasionally but only with selected drug examples, and its relevance for brain permeation remains elusive for the non-specialist reader (some agonists are pretty big structures, see Fig. 5, and may be PGP substrates). There are other pharmacological phenomena which are poorly explained, e.g. spare receptors. Maybe a brief discussion of some of these points in a separate box would improve reader understanding.

The authors appreciate the reviewer’s comments suggesting a further discussion of important points relating to pharmacology. We have added a brief mention of how the Kpu,u value was calculated for HTL18138 and what is currently thought to be a ‘good’ value for sufficient BBB permeability in a text box/footer section. Additionally, we make mention of spare receptors/receptor reserve in this same text box/footer (we will confirm with our Assistant Editor that this formatting is appropriate before submission) as the concept is initially mentioned in line 235. Finally, we will add a brief explanation of PGP in the text around the section of the manuscript where it is first mentioned (line 343). We will further clarify its relevance in drug development and describe how it is addressed with brain/plasma measurement.

2. Another phenomenon that is not well discussed is the fact that acetylcholine, the physiologic agonist, is a permanent cation, and that it is difficult to model an orthosteric ligand that penetrates the blood-brain barrier and has agonistic action on receptors that are used to interacting with a cationic transmitter.

The reviewer raises an important point. A brief mention of the necessity of protonation of the hindered amine to mimic ACh has been added to lines 144-146 in addition to a second mention in the concluding paragraph of the section (lines 268-271) regarding the challenge of designing/modeling an orthosteric agonist to mimic the cationic nature of ACh.

3. Finally, from a pharmacological standpoint, continuous activation by agonists is clearly inferior to increasing the “cholinergic tone” e.g. by inhibiting AChE or using PAMs. This point is made by the authors but sort of hidden away among the lengthy drug discussion.

The authors appreciate the reviewer pointing out this detail. Indeed, direct-acting agonists appear to be inferior to enhancing cholinergic tone via PAMs or AChEIs—albeit a few recent examples of functionally selective mAChR agonists, which are detailed in the review. However, while AChEIs (e.g., tacrine and donepezil) have demonstrated dose-dependent efficacy in improving cognition in patients with early-stage AD, they suffer from dose-limiting side effects attributed to non-selective activation of cholinergic receptors in the CNS and periphery, which limits clinical utility (noted in lines 116-122). However, we have added this more explicitly in line 300 (e.g., cholinergic tone) and in lines 479-481 in the concluding paragraph of section 2.2.

Further comments

4. Line 424-425, please move this sentence to the end of the paragraph, after the discussion on schizophrenia. AD patients do not have cognitive dysfunctions below 30 years of age.

The authors appreciate the reviewer's comments regarding how the paragraph appears to read that AD symptoms begin below 30 years of age, which is not correct. Lines 424-425 have been moved to the end of the paragraph (now lines 433-436), along with minor grammatical changes. Corresponding references were also corrected to reflect this shift in the text.

5. The text in lines 898-904 is identical to lines 905-910, and the figure should be 9, not 8. 

The authors appreciate the reviewer finding this repeating text. It has been removed and Figure 8 in line 1005 (in the revised document) has been changed to Figure 9.

Minor comments

6. Line 79, delete “was present”. 

This has been deleted. 

7. On lines 215-220, one wonders why a press release is cited but then the results were not “made public”: Do the authors mean that there was no publication with detailed information?

The authors appreciated this being brought to our attention, and we understand how this could be perplexing. We have clarified this to say that was no publication of this detailed information by the company.  

8. Line 376 ff., it is somewhat surprising that compounds that were once developed for Alzheimer´s dementia (cholinergic hypofunction) were later repurposed to Parkinson´s disease (which is hypercholinergic in a way). 

Indeed. The authors appreciate the reviewer raising this point. This is likely due to ANAVEX 2-73's activity at the sigma 1 receptor. By activating sigma 1, it could be expected to lower the toxic accumulation of misfolded proteins in nerve cells, dysfunction in mitochondria, as well as oxidative stress and neuroinflammation — which are all involved in Parkinson’s, AD, and Rett syndrome. As readers may raise a similar question, we have noted this in the text. 

9. Line 595: oxotremorine is non-selective in vitro, but more or less M2-specific in vivo (talk about spare receptors…). 

The authors appreciate this point being brought up regarding spare receptors ('receptor reserve’ or ‘signal amplification'- as described in Hill et al., 2010; Rajagopal et al., 2011), and it is indeed an important topic to discuss. However, the authors believe a brief discussion may be better placed in the section of the review where receptor reserve is initially mentioned (lines 234-235). This will be added in the revised manuscript with appropriate sources. 

10. Line 812, do these effects of scopolamine (antidepressant and anti-cognitive) occur at the same doses?

The authors appreciate the reviewer raising this question. In short, the dosages producing antidepressant effects vs. anti-cognitive effects can be distinguished. We have addressed this briefly at the end of line 812 with appropriate sources. 

11. Line 822, scopolamine has a short clearance time from plasma but has a long residence in time in most tissues in the body, including the brain.

The authors appreciate the reviewer mentioning scopolamine’s extended residence time in most tissues (including the brain). A brief note will be added directly below this line to indicate this in the revised copy of the manuscript.

12. Line 823, should be “blockade”, not “activation”.  

The authors appreciate the reviewer finding this error. It has been corrected to "blockade."

Reviewer 2 Report

In the present review, the Authors have highlighted the historical and recent drug development efforts and design, specifically for Alzheimer’s disease (AD), schizophrenia, and major depressive disorder (MDD). In addition, although not exhaustive, some of the most promising compounds and chemical scaffolds were discussed to illustrate their utility in treating these pathologies. 

Overall, I found this review timely, original, well written and scientifically sound. I have only some minor suggestions aimed to improve the high quality of the paper and these are outlined below:

1) How the literature searches were conducted and relevant articles selected and included in the review? Please, add a very brief note on this point.

2) As I found the present narrative review very interesting, I guess why the Authors didn't conducted a systematic review.

3) I agree with the Authors when they in page 11 line 450 wrote that "However, it should be noted that the dopamine hypothesis of schizophrenia alone cannot account for the wide range of positive, negative, and cognitive symptoms experienced by patients...". There are intriguing hypotheses concerning the role of postsynaptic density and its relationships with the glutamatergic system in schizophrenia. Please add a brief comment on this point with appropriate references (see dois: 10.3390/ijms18010135 and 10.2174/1568026621666210701103147). 

Author Response

**Please note that the lines mentioned in response to this review reflect the current revised manuscript (with tracked changes), and therefore may not match the original lines referenced by the reviewer.**

1) How the literature searches were conducted and relevant articles selected and included in the review? Please, add a very brief note on this point.

Literature searches were conducted through standard databases (Scifinder, Pubmed, Google, etc. among other institutional resources). Many of the sources were already known to the authors previously as they have studied the mAChRs in detail. A chronologically ordered approach was taken, starting from the origination of the cholinergic hypothesis, followed by Merck’s interest in mAChR agonists in the treatment of AD, then shifting to allosteric modulation (and improved design of select orthosteric compounds). The authors have added a brief note to the introduction paragraph (begin line 75) summarizing the above.

2) As I found the present narrative review very interesting, I guess why the Authors didn't conduct a systematic review.

The authors are glad to hear that the reviewer found this literature review timely, relevant, and interesting. We did not conduct a systematic review as the goal was not to answer a specific and focused clinical question, but rather to provide a broad overview (i.e., literature review) of drug design targeting the mAChRs in AD, SZ, and depression. 

3) I agree with the Authors when they in page 11 line 450 wrote that "However, it should be noted that the dopamine hypothesis of schizophrenia alone cannot account for the wide range of positive, negative, and cognitive symptoms experienced by patients...". There are intriguing hypotheses concerning the role of postsynaptic density and its relationships with the glutamatergic system in schizophrenia. Please add a brief comment on this point with appropriate references (see dois: 10.3390/ijms18010135 and 10.2174/1568026621666210701103147). 

The authors appreciate the reviewer’s comments on the topic of PSD and its relationship to glutamatergic signaling. We have added several sentences directly under (lines 530-531) to further describe this post-synaptic cluster of proteins located at excitatory glutamatergic post-synapses with appropriate citations. Also, the following sentence (next paragraph) further acknowledges the involvement of the glutamatergic pathway in SZ when it refers to the “psychotomimetic effects of NMDA/mAChR antagonists….”. While only a brief summary, a similar point is mentioned in 10.2174/1568026621666210701103147 specifically referring to PCP and ketamine.